# Gut Microbiota, Macrophages and Diet: An Intriguing New Triangle in Intestinal Fibrosis

**DOI:** 10.3390/microorganisms10030490

**Published:** 2022-02-22

**Authors:** Asma Amamou, Cian O’Mahony, Mathilde Leboutte, Guillaume Savoye, Subrata Ghosh, Rachel Marion-Letellier

**Affiliations:** 1APC Microbiome Ireland, College of Medicine and Health, University College Cork, T12 YT20 Cork, Ireland; 121126891@umail.ucc.ie (C.O.); subrataghosh@ucc.ie (S.G.); 2INSERM UMR 1073 “Nutrition, Inflammation and Gut-Brain Axis”, Normandy University, 76183 Rouen, France; mathilde.leboutte@etu.univ-rouen.fr (M.L.); rachel.letellier@univ-rouen.fr (R.M.-L.); 3Department of Gastroenterology, Rouen University Hospital, 76031 Rouen, France; guillaume.savoye@chu-rouen.fr

**Keywords:** inflammatory bowel diseases, fibrosis, macrophages, gut microbiota, diet

## Abstract

Intestinal fibrosis is a common complication in inflammatory bowel disease (IBD) without specific treatment. As macrophages are the key actors in inflammatory responses and the wound healing process, they have been extensively studied in chronic diseases these past decades. By their exceptional ability to integrate diverse stimuli in their surrounding environment, macrophages display a multitude of phenotypes to underpin a broad spectrum of functions, from the initiation to the resolution of inflammation following injury. The hypothesis that distinct macrophage subtypes could be involved in fibrogenesis and wound healing is emerging and could open up new therapeutic perspectives in the treatment of intestinal fibrosis. Gut microbiota and diet are two key factors capable of modifying intestinal macrophage profiles, shaping their specific function. Defects in macrophage polarisation, inadequate dietary habits, and alteration of microbiota composition may contribute to the development of intestinal fibrosis. In this review, we describe the intriguing triangle between intestinal macrophages, diet, and gut microbiota in homeostasis and how the perturbation of this discreet balance may lead to a pro-fibrotic environment and influence fibrogenesis in the gut.

## 1. Introduction

Inflammatory bowel diseases (IBD) are chronic relapsing and remitting inflammatory diseases affecting the gastrointestinal tract, resulting from complex interactions between genes, environment exposure, especially food, the immune system, and gut microbiota. IBD are often progressive and lifelong pathologies leading to irreversible intestinal damage causing disability and morbidity in patients [1]. Among these complications, intestinal fibrosis is the most common. Indeed, intestinal fibrosis affects up to 50% of Crohn’s disease patients (CD) and to a lesser degree up to 11% of ulcerative colitis (UC) patients [1]. Intestinal fibrosis is characterised by exaggerated scar tissue formation in intestinal mucosa resulting from chronic inflammation. In IBD, while specific targeted therapies have made a major impact on inflammation, the incidence of fibrostenotic complications has not substantially reduced and the development of new therapies is urgently needed [2]. About 80% of patients undergo surgical resection with loss of viable tissue. [3,4]. However, in up to 70% of patients, ulcerations and strictures commonly recur, and repeated resections are often necessary with significant impacts on the patients’ physical and mental wellbeing [1,5]. Mechanisms orchestrating intestinal fibrosis require further insight to enable therapies that may alter the course of this complication. In this review, we highlight the role of macrophages in intestinal fibrogenesis and their interactions with intestinal microbiota and diet and discuss the potential future directions of treatments by exploiting macrophage function and plasticity.

## 2. Macrophages as Key Players in Fibrogenesis

The gut mucosa hosts the largest population of macrophages in our organism. These intestinal macrophages play pivotal roles in the maintenance of intestinal homeostasis. Although macrophages are effector cells of the innate immune system, they are also involved in mucosal healing, angiogenesis, and tissue metabolism [6]. Macrophages are present in all the phases of mucosal healing exhibiting distinct phenotypes associated with specific tasks [7]. Commonly, macrophages, referred to as pro-inflammatory phenotype M1, are the first macrophages in place following an injury to clean the local environment of a wound from bacteria, dead cells, and debris [8]. During the initiation of tissue repair, macrophages undergo phenotype change into anti-inflammatory and pro-regenerative macrophages, referred to as M2, which produce key mediators, such as transforming growth factor β1 (TGF-β1), to promote the migration and differentiation of tissue fibroblasts which produce extracellular matrix (ECM) components that facilitate wound healing [7]. Finally, during the remodelling phase, macrophages release matrix metalloproteinases (MMP), restoring tissue integrity [9]. Thus, wound healing is a complex, tightly regulated process; however, when injuries are repeated, this process persists and becomes uncontrolled, leading to complications, especially fibrosis [10]. The functional plasticity of macrophages depends on adaptation to their tissue environment, changes which provoke reprogramming of their metabolism and the induction of a complex gene program regulated by numerous transcription factor families [11]. The composition of the local tissue microenvironment, such as cytokines, growth factors, microbial products, along with other immune and stromal cells, are critical determinants of the macrophage responses. Over the last decade it has become clear that macrophage function in tissue repair is complex and contributes to several chronic diseases [12]. Considering their critical role in the inflammatory response and mucosal healing, macrophages have generated considerable interest in deciphering their involvement in chronic inflammatory disorders, including IBD [13,14]. IBD patients and experimental models of colitis show an accumulation of macrophages in inflamed tissue [15,16]. It is now well established that M1 macrophages play a critical role in the chronicity of inflammation associated with IBD [15]. Interestingly, many recent studies have demonstrated the involvement of macrophages in fibrogenesis of the lungs, kidneys, and pancreas [17,18,19]. The contribution of macrophages to intestinal fibrogenesis remains poorly studied; however, because similarity in histomorphology is observed in fibrosis involving all organs, some of the mechanisms underpinning fibrogenesis are likely to be common to all these organs, including in the gastrointestinal tract.

In the lungs, liver, and kidneys, a co-localization of macrophages and myofibroblasts has been observed at fibrotic sites [20,21,22]. These studies highlight the importance of the interaction between macrophages and myofibroblasts in the fibrotic tissue. Proliferation and differentiation of myofibroblasts are the central events in the fibrogenesis process [23]. While multiple factors influence the promotion of myofibroblast expansion, TGF-β1 remains the most critical in fibrosis progression [24]. TGF-β1 mediates progressive intestinal fibrosis in both experimental and human IBD through Smad2/Smad3 pathway activation by stimulating ECM production whilst inhibiting its degradation. Moreover, TGF-β1 acts also by inducing the transformation of both epithelial and endothelial cells to myofibroblasts through epithelial–mesenchymal transition (EMT) and endothelial–mesenchymal transition (EndoMT) [25]. In extra-intestinal fibrosis, several studies highlighted the critical role of macrophages, especially M2 subsets, as a critical source of TGF-β1 [26]. In liver fibrosis, TGF-β1 has been described as having a dual antagonistic role. Indeed, TGF-β1-producing regulatory T (Treg) cells have been shown to decrease fibrosis through an interleukin (IL)-10-dependent mechanism, whereas macrophage-derived TGF-β1 exerts pro-fibrotic effects [27]. As a matter of fact, many studies have identified several macrophage subtypes as essential producers of TGF-β1 with the profibrotic cytokine IL-13, which is able to induce and activate latent TGF-β1 through an MMP9-dependent mechanism [28,29]. Altogether, this suggests that TGF-β1 activity inhibits inflammation if produced by Treg cells but promotes fibrosis when produced by macrophages. The fact that TGF-β1 can simultaneously suppress inflammation but promote collagen synthesis in myofibroblasts likely explains the dual role of this cytokine. Targeting the inhibition of the specific profibrotic macrophage, rather than globally alleviating TGF-β1, might provide a more rational approach to ameliorate fibrosis. Indeed, studies have already shown that decreasing the number of TGF-β1-producing macrophages significantly slows the progression of hepatic fibrosis [30]. Production of TGF-β1 is mainly conducted by M2 macrophages. During fibrogenesis of the heart and kidneys, disturbance to the balance M1/M2 macrophages have been observed, with a switch of M1 towards M2 [31,32]. In unilateral ureteral obstruction, an experimental model of renal fibrosis, higher levels of TGF-β1 were detected in M2 macrophages than in myofibroblasts [32]. In this study, the invalidation of STAT6, a key transcription factor in M2 differentiation, decreased the production of collagen and the activation of TGF-β1/Smad signalling and inhibited renal fibrosis.

M2 macrophages can be further classified into four subtypes according to their patterns of biomarkers: M2a, M2b, M2c, and M2d. These macrophage subsets play distinct roles in the inflammatory response and wound-healing process (Figure 1). However, factors influencing subset polarisation, and their specific functions in vivo remain unclear. Nonetheless, in vitro, M2a macrophages are typically induced by IL-4 and IL-13; the M2b macrophages are induced in response to immune complexes and bacterial lipopolysaccharide (LPS), and the M2c macrophages to glucocorticoids, IL-10, and TGF-β. Lastly, the M2d macrophages are induced in response to IL-6 and adenosines [33]. Recent studies suggest that specific M2 subpopulations could be assigned to fibrogenesis progression while certain others are assigned to wound repair (Figure 1). In renal fibrosis, the roles of macrophages M2a and M2c have been recently described [34]. In a murine model of renal fibrosis, an increased M2a infiltration is observed and correlated with a higher fibrosis score [34]. In this study, the inhibition of M2a macrophage infiltration was inhibited following administration of trichostatin A, a histone deacetylase (HDAC) inhibitor. Accordingly, an upregulation of M2c macrophage populations in the kidneys was observed and was associated with a diminution of renal fibrosis [34]. Indeed, M2a macrophages produce pro-fibrotic factors, including TGF-β1, connective tissue growth factor (CTGF), fibroblast growth factor (FGF), and insulin-like growth factor (IGF) [35]. M2c macrophages act partly via STAT1 and NF-kB inhibition, leading to inactivation of myofibroblasts and restriction of M1 and M2a macrophage subsets via a local production of IL-10 [35]. In line with these observations, in cardiac fibrosis, the different M2 macrophage phenotypes have divergent effects on cardiac fibroblasts, especially the M2a (pro-fibrotic) and M2b (anti-fibrotic) phenotypes. Indeed, the intracardiac injection of M2b macrophages in a rat model of cardiac ischemia/reperfusion injury significantly suppressed the proliferation and migration of cardiac fibroblasts. Moreover, this approach attenuated the expression of fibrosis-related proteins and abrogated the differentiation of cardiac fibroblasts into myofibroblasts [36]. These studies suggest that M2a macrophages are profibrotic and could be critically involved in fibrogenesis. Thus, the dual role of M2 macrophages could be explained by the balance of distinct M2 subpopulations in chronic fibrotic disorders.

In intestinal fibrosis, the roles of M2 macrophages are not yet well elucidated. However, recent studies using an experimental rat model of IBD underlined the critical role of M2 macrophages in the progression of intestinal fibrosis and its associated complications, such as strictures. In the TNBS rat model, an important M2 phenotype of macrophages was specifically detected in the underlying smooth muscle at the stricture site [37]. In this study, the authors developed an in vitro model of the stricture phenotype using intestinal smooth muscle cells and showed that TGF-β1 secreted by M2 macrophages led to a hyperproliferative phenotype of these cells, whilst improving their survival in hypoxic conditions. In the dextran sulfate sodium (DSS)-induced colitis model, DSS administration directs the phenotype of macrophages towards the M2 lineage and seems to inhibit pathogen clearance and exacerbate inflammation, contributing to fibrosis development [38]. However, several other studies report the beneficial effects mediated by M2 macrophages. M2b macrophage-derived exosomes reduced the severity of DSS-induced colitis and promoted mucosal healing in mice [39]. These contradictory results may be explained by a disrupted balance of distinct M2 subpopulations in IBD. Thus, promoting the reprogramming of macrophages to an “anti-fibrotic M2 subset”, such as M2b, could be a potential strategy in the treatment of fibrogenesis, including intestinal fibrosis.

In intestinal biopsies from CD patients, M2 macrophages are found in fibrotic areas, but the causal effect of M2 macrophages and its subtypes in intestinal fibrosis remain unrevealed [40]. However, some lessons from extra-intestinal fibrosis, such as cardiac fibrosis, could be partially true for the gut. It is essential to emphasise that M2 macrophages assume a spectrum of phenotypes with different functions, whose equilibrium during the disease are key events for recovery. Thus, studies in which M2 macrophages are found to be beneficial for fibrosis may refer to an increase of antifibrotic activity and an alleviation of pro-fibrotic activity. That is why it appears essential to unravel the complexity behind the macrophage polarisation and most particularly the M2 subtypes which promote wound healing and counteract intestinal fibrogenesis.

## 3. Factors That Influence Macrophage Polarisation in Inflammation and Fibrosis

Polarisation of macrophages is regulated by several factors. Among cellular modulators, innate immunes cells are the most reported [41]. Indeed, the adaptive immune response through T cells could influence the expansion and function of macrophages. Th1 cells and associated cytokines (IFNγ, IL-2, TNF-α) are known to promote M1 macrophage polarisation, while Th2 cells and associated cytokines (IL-4 and IL-13) promote M2 macrophages [41]. However, macrophage subtypes subsist without adaptative cells, including B and T cells [42]. Over the last decade, two key factors emerged as relevant to immune response dysfunction: gut microbiota and diet.

### 3.1. Gut Microbiota

It is increasingly recognised that gut microbiota influences host immune cell fate, including macrophages, and leads to chronic inflammation, carcinogenesis, or fibrosis. The gastrointestinal tract harbours the largest community of microorganisms, estimated at over 100 trillion microbes and more than 500 different species [43]. The fundamental role of gut microbiota in the development and functions of the host was highlighted several decades ago. Most of these studies have established that the intestinal microbiota maintains a tolerogenic environment via the induction of M2 intestinal macrophages [44]. Interplay between these microbes and macrophages along the gut microbiota–host axis is critical in the maintenance of intestinal homeostasis. Accordingly, several reports have demonstrated the implication of this axis in the pathogenesis of chronic disease, including IBD [14].

In the intestine, the lamina propria contains a large number of macrophages playing a key role in sensing and killing invading microbes, eliminating dead cells, and contributing to wound healing and epithelial repair [40]. Studies using either germ-free or specific pathogen-free mice have reported the essential role of gut microbiota in modulating macrophage diversity in the colon. Moreover, these effects extend to regulating macrophage function and development from precursors in the bone marrow [40,45]. Several bodies of evidence have described that gut microbiota shape intestinal macrophage via direct or indirect pathways. For example, *Bacteroides fragilis* and *Clostridia* class induce M2 polarisation [46]. In contrast, some bacteria, such as *Enterococcus faecalis*, polarise colonic macrophages toward the proinflammatory M1 phenotype in mice [47].

The preponderant role of gut bacteria in the pathogenesis of IBD has been highlighted by both human and animal studies. In addition, the effects of microbial products, such as short chain fatty acids (SCFAs), on the immune response in IBD are well characterised. Hayashi et al., demonstrated that a *Clostridium butyricum* probiotic strain decreases inflammation stemming from DSS-induced colitis in mice. Intriguingly, this mechanism was shown to act independently of T cell signalling [48]. In this study, *Clostridium butyricum* directly induced IL-10 production by intestinal macrophages in inflamed mucosa and triggered polarisation of the macrophages into the anti-inflammatory phenotype through the TLR-2/MyD88 pathway. In addition, SCFAs depletion induced by antibiotics favours an M1 hyper-responsive phenotype, leading to an overproduction of pro-inflammatory cytokines.

Up to now, particular attention has focused on the crosstalk between microbes and inflammatory signalling in IBD. However, few studies have investigated the role of the gut microbiota in intestinal fibrosis. The lipopolysaccharide (LPS), a component of the outer membranes of Gram-negative bacteria, is a type of pathogen-associated molecular pattern (PAMP) proven to be pro-fibrotic. Indeed, in vitro stimulation of intestinal fibroblasts with LPS, leads to enhanced TGF-β1/Smad signalling and increased ECM protein secretion through TL4 [49]. Direct profibrotic effects of bacteria have been observed in murine models. For example, *Clostridium innocuum* is able to translocate from the gut lumen to mesenteric adipose tissue (MAT) in surgically resected samples from CD patients [50]. This was further confirmed in DSS-treated altered Schaedler flora (ASF) mice where *C. innocuum* bacteria were found in the MAT and selectively activated pro-fibrotic macrophages in chronically inflamed states [51]. Activated macrophages released various cytokines and growth factors, leading to mesenteric adipose tissue expansion and intestinal fibrogenesis [50]. However, in these studies the characterisation of macrophage subpopulations induced by gut microbiota leading to fibrogenesis remain largely unexplored. In intestinal inflammation caused by *Fusobacterium nucleatum*, the selective recruitment of M2-like macrophages via the contribution of TLR4 and the subsequent activation of the IL-6/STAT3 pathway is observed and M2 macrophage polarisation was denoted by heightened expression of the M2 marker, TGF-β1 [52]. Darfeuille-Michaud et al., have first identified a specific bacterial pathotype of *Escherichia coli* called *adherent-invasive Escherichia coli* (AIEC) that is also able to colonise the intestinal epithelium [53]. In this study, the authors also demonstrated the ability of AIEC to survive and replicate within macrophages without inducing cell death [53]. This AIEC pathotype even develops a resistance mechanism to phagolysosomal conditions [54]. Consequently, persistent colonisation by AIEC led to colonic fibrosis in mice. The induction of intestinal fibrosis was mediated by flagellin, a principal component of bacterial flagella, which activated IL-33 signalling. IL-33 signalling induces the activation and the differentiation of TGF-β1-producing cells, such as M2 macrophages, and subsequently causes the development of fibrosis [55]. In line with this observation, in irradiation-induced intestinal fibrosis in rats, the authors observed a progressive shift of colonic macrophages towards the M2 phenotype driven by flagellin [56]. Indeed, several studies have reported that LPS and flagellin can both promote macrophage polarisation towards the M2 phenotype [49,57].

Nevertheless, mechanisms by which gut microbiota regulate macrophage reprogramming during intestinal fibrosis remain unknown. Therefore, identifying the key “switches” that drive macrophage polarisation could lead to the development of an effective therapeutic approach for fibrosis related to chronic disease, including IBD.

### 3.2. Diet

It is now evident that many dietary components could influence the polarisation of immune cells, including macrophages, and alter the progression of chronic inflammatory disorders [58]. Fatty acids, such as conjugated linoleic acid (CLA), have been studied extensively for their ability to modulate the inflammatory response. CLA has been shown to inhibit expression of pro-inflammatory cytokines, such as IFN-γ, and induce the polarisation of macrophages to the M2 phenotype in the mouse macrophage cell line RAW264 [59]. The underlying mechanism of CLA action was shown to involve the binding of CLA to the PPAR-γ receptor.

PPAR-γ is a ligand-activated nuclear receptor which plays a key role in the differentiation and polarisation of macrophages. Previous studies have documented that PPAR-γ ligands alleviate inflammatory responses, including repression of nuclear factor-kappa B (NF-κB) and signal transducer and activator of transcription (STAT) signalling [60]. PPAR-γ, largely expressed in macrophages, is induced by several natural and synthetic ligands [61]. Notably, curcumin, a dietarily derived polyphenol, is a natural agonist of PPAR-γ. In vitro, curcumin increased expression of M2 markers, such as the mannose receptor and arginase-1, in a human monocyte THP1 cell line and promoted the secretion of IL-4 and/or IL-13 [62]. Besides possessing anti-inflammatory properties, curcumin also exhibits anti-fibrotic effects in several organs, which are attributed to PPAR-γ induction. In bleomycin-induced lung fibrosis in mice, curcumin inhibited TGF-β1/Smad signalling and decreased the proliferation and migration of fibroblasts. Beneficial effects of curcumin on intestinal fibrosis in IBD models have been demonstrated [63]. In a chronic TNBS-induced colitis model, curcumin exerted anti-fibrotic effects on a rat intestine. These effects were ascertained to be dependent on PPAR-γ activation and subsequent inhibition of the TGF-β/Smad signalling-mediated EMT [64]. However, the effects on macrophage responses were not studied and should be considered in future investigations.

Glutamine is the major dietary amino acid which has been extensively studied for its immunomodulatory effects in several pathologies, including IBD [65]. Indeed, glutamine reduces pro-inflammatory signalling and promotes M2 macrophage polarisation through uridine 5′-diphosphate-nacetylglucosamine biosynthesis and N-glycosylation [66]. Furthermore, evidence of anti-fibrotic effects of glutamine supplementation has been observed in several organs. In murine carbon tetrachloride (CCl4)-induced liver fibrosis, glutamine supplementation inhibits TGF-β1 induced-EMT and reduces collagen deposition in mouse hepatocytes [67]. Accordingly, glutamine supplementation reduced ECM-associated protein deposition in colonic submucosa of TNBS-induced colitis rats [68]. However, despite these interesting protective effects, glutamine has never been specifically evaluated in IBD patients with intestinal fibrosis. In addition, recent observations suggest that glutamine could drive fibrosis in the lungs by producing proline and glycine, the two most abundant amino acids constituting collagen protein [69]. Considering the involvement of glutamine in M2 macrophage polarisation and some adverse effects of this amino acid in fibrosis, a deeper understanding of the relationship between glutamine, macrophages, and fibroblasts is required to strongly consider glutamine supplementation from a therapeutic perspective.

Flavonoids are derived from plant metabolites and are mainly present in plant-based foods. Flavonoids represent an interesting class of nutritional compounds and have been investigated for their possible anti-inflammatory activity. Indeed, in a murine model of non-alcoholic steatohepatitis (NASH) induced by the choline-deficient, L-amino acid-defined, high-fat diet (CDAHFD), administration of myricetin, a plant-derived flavonoid, induced an M2 polarity switch in liver macrophages [70].

In contrast, diet can also negatively affect the immune system, contributing to the emergence of chronic inflammatory diseases. In our industrialised society, immune-mediated diseases are increasing, and among environmental factors, our diet has been gaining attention. The introduction of the western diet, high in fat and protein and low in fruits and vegetables, has been implicated in the development of several chronic disorders. Evidence suggests that these deleterious effects are driven by macrophages. In a high-fat diet (HFD)-induced obesity murine model, it has been demonstrated that the expression of the M2 macrophage markers dectin-1, CD36, and arginase 1 is greatly increased. Expression of these markers is characteristic of M2b macrophage polarisation, which in turn participates in chronic inflammation [71]. A similar switch is observed in the adipose tissue of obese patients [72]. These observations reinforce the concept that metabolic abnormalities induced by diet are able to polarise macrophages, whatever their tissular origin, towards an M2 phenotype. Consequences of HFD have been studied in experimental models of colitis. In TNBS-induced colitis, HFD feeding exacerbated colitis in mice and is associated with an alteration of immune response, reflected by higher expression of mucosal pro-inflammatory cytokines. Moreover, extensive colonic injury was observed, with a higher degree of mucosal ulceration [73]. A diet high in saturated fat increases the risk of developing IBD [74]. Accordingly, several studies highlight the importance of the ratio n-3:n-6 polyunsaturated fatty acids (PUFAs) in inflammatory disease, including IBD [75,76]. N-6 PUFAs are pro-inflammatory, while n-3 PUFAs are anti-inflammatory and are capable of countering the effect of n-6 PUFAs. It has been shown that anti-inflammatory proprieties of n-3 PUFAs are underpinned by the polarisation of macrophages towards an M2 phenotype, therefore regulating the onset of HFD-induced tissue inflammation and metabolic perturbations [77]. Interestingly, n-3 PUFA are shown to exhibit anti-fibrotic effects by inhibiting M2 macrophage activation and TGF-β1 signalling in peritoneal fibrosis in mice [78]. Once again, these studies exhibit the dual role of M2 macrophages during inflammation and fibrosis. These controversial observations could be explained by the M2 subpopulations and their distinct role in wound healing and fibrogenesis.

Salt is another critical dietary component positively associated with risk for many diseases which could affect the polarisation of macrophages. Indeed, the direct effects of salt on immune cells, such as macrophages, have been demonstrated in several experimental studies. The effects of a high salt diet are well characterised in cardiovascular and kidney diseases. In vitro and in vivo, high salt concentrations (up to 250 mM) significantly reduced the activation of M2 macrophages (producing IL-4 and IL-13) and increased M1 macrophages (producing IFN-γ) [79]. A high-salt diet is also involved in fibrogenesis. In a recent study, high salt-exposed macrophages showed significantly increased surface markers encoding for M2-type macrophages, such as mannose receptor and arginase 1 [80]. In vitro macrophages co-cultured with renal fibroblasts and exposed to high levels of salt exhibited significantly increased proliferation and migration ability. Moreover, protein expression of collagen I, collagen III, and α-SMA in renal fibroblasts is greatly enhanced. High-salt exposure induces polarisation of mononuclear macrophages to the M2 macrophage phenotype and induces secretion of both IL-6 and TGF-β1. Subsequently, these cytokines induce the proliferation and phenotypic transformation of renal fibroblast cells [80]. In IBD, the effect of dietary salt is poorly documented, but a recent retrospective study from two European cohorts (129 IBD patients, 56 controls) found a higher consumption of salt in IBD patients (OR 2.8 [1.5, 5.3]) [81]. Accordingly, a few studies using murine colitis models have observed that a high-salt diet exacerbates colitis, inducing an environment increasingly vulnerable to inflammatory insults [82]. Recently, we have characterised the effect of a high-salt diet on intestinal fibrogenesis using murine chronic colitis and cellular models of intestinal fibrosis [83]. Indeed, in our study, a high-salt diet increased mRNA transcripts encoding for pro-fibrotic markers, such as CTGF and collagen I, in chronic TNBS-induced colitis rats. In addition, in TGF-β1-induced human colon fibroblasts, we found that NaCl induced ECM-associated proteins. These results suggested that dietary salt could be contributing to intestinal fibrosis progression. Additional clinical trials are needed to investigate whether dietary salt influences the natural history of IBD and to characterise the role of macrophages in this process.

## 4. Interplay between Diet, Gut Microbiota, and Macrophages in Intestinal Fibrosis

Gut microbiota composition and function are dependent on several factors, including diet. Investigations of the interactions between food and gut microbiota and the ensuing impact on the health of the host have gathered increasing interest over these last decades. Various perturbations to intestinal microbiota have been detected in different chronic inflammatory diseases, including IBD [84]. However, results have been heterogeneous and the mechanisms by which intestinal microbiota regulate fibrotic processes remain unrevealed. As mentioned above, the production of metabolites by gut microbiota can shape immune cell responses, including macrophages. It is now clear that diet could drastically modulate luminal microbiota-derived metabolite concentrations by affecting substrate availability in the gut microenvironment.

Microbiota-derived metabolites could reprogram the macrophage immunometabolism pathway to modulate host immunity and lead to inflammatory disorders [85]. Indeed, evidence regarding the metabolic shift dictated by TLR signalling is well established but understanding of how these metabolic pathways interact with gene expression and macrophage activation remains limited.

Macrophages upregulate glycolysis to support acute energy production and facilitate rapid pro-inflammatory responses, whereas oxidative phosphorylation and fatty acid oxidation provide sustained energy for macrophage anti-inflammatory functions [85]. Overall, macrophages are capable of sensing their surrounding metabolic environment, and nutrient provision can modulate macrophage function. The gut microbiota modulates these processes, notably through production of SCFAs. SCFAs are the main product of dietary fibre fermentation by gut microbiota. Consumption of a high-fibre diet promotes the expansion of beneficial bacteria species, such as *Lactobacillus* and *Bifidobactria* [86]. Butyrate or propionate produced by gut microbiota reprogram macrophage metabolism toward lipid oxidation, leading to polarisation of macrophages towards the anti-inflammatory M2 phenotype [87]. In mice, treatment with antibiotics diminishes production of butyrate by gut microbiota and leads to a polarisation of intestinal macrophages towards M1 macrophages [88]. In IBD, the populations of commensal bacteria (predominantly *Firmicutes* and *Bacteroidetes*) are reduced [84]. Furthermore, decreased faecal concentrations of SCFAs have been identified, along with reductions in the populations of the main butyrate-producing microbes, the *Clostridium IXa* and *IV* groups. Understandably, these fluctuations have been associated with the accumulation of pro-inflammatory macrophages [89].

Accordingly, in vivo administration of butyrate attenuates gut inflammation and lesions in both IBD patients and rodent models [90]. Thus, regulation of metabolic process occurring in intestinal macrophages is crucial in IBD pathogenesis and could influence fibrogenesis.

Dietary saturated fatty acids (SFAs) are common in foods, such as palm oil, beef, pork, or cheese, and can promote pro-inflammatory effects. These are associated with alterations to the gut microbiota and are characterised by a diminution of *Bacteroides* species [91]. Interestingly, the effects of the SFAs palmitic acid on macrophages have been recently identified. Namely, through enhanced TLR4 activation, palmitic acid promotes the expression of the transcription factor NF-κB, which induces pro-inflammatory factors, such as COX-2, TNFα, IL-1β, IL-6, CXCL8, IL-12, and IFN-γ in macrophages [87]. Moreover, in a murine model of NASH, dietary palmitic acid promotes liver fibrosis by activation of TGF-β1-producing cells, including macrophages and hepatic stellate cells [92].

Malnutrition and micronutrient deficiencies are common in IBD patients [93]. Among these, zinc deficiency is found in up to 14% of IBD patients [94] and has been associated with a poorer clinical outcome in these patients [95]. Existing data demonstrate a close link between zinc host metabolism and gut microbiota. Indeed, zinc participates in several bacterial processes, including replication and survival. That is why shutting zinc off from disease-causing bacteria is a common strategy used by the immune system to prevent their growth [96]. Recent studies shows that dietary zinc is a critical factor for gut bacteria biodiversity [96]. Similarly, the modulation of gut bacteria is considered a key factor in the regulatory role of zinc in immunity. Indeed, Gordon et al., provide evidence of Th17 cell modulation in the intestines of mice through the gut microbiome after zinc supplementation [97]. Of note, zinc deficiency exacerbates murine colitis by increasing the proportion of M1 macrophages in experimental IBD models [98]. Interestingly, dietary zinc supplementation can reduce extra-intestinal (i.e., liver) fibrosis in mice through selective suppression of M1 macrophages [99]. It is therefore tempting to speculate that dietary zinc supplementation could alleviate intestinal fibrosis in IBD.

Vitamins are also a key substrate for macrophages. While vitamins are extracted from the diet, intestinal microbiota possess a critical role in their production and bioavailability. Therefore, a dysfunctional microbiota could propagate impaired vitamin synthesis and metabolism and could participate in the pathogenesis of several inflammatory diseases [100]. For example, severe obesity is associated with gut dysbiosis and a recent study by Clément et al., has demonstrated the link between gut microbiota, vitamin B, and metabolic response [101]. Indeed, patients with severe obesity exhibited a deficiency in bacteria involved in the production and/or transport of biotin, leading to reduced circulating biotin levels, and the abundance of these bacterial species are correlated with host metabolism and inflammatory response [101]. The role of fat wrapping of the intestine in response to bacterial translocation through micro-penetrating disease with involvement of macrophages and adipocytes has been recently highlighted in human disease and animal models [50]. Vitamin D deficiency has been observed in patients with IBD. Moreover, the importance of vitamin D in intestinal homeostasis has been highlighted by previous studies, which have shown that, in IBD, vitamin D is intimately involved in the regulation of inflammatory macrophages via a bidirectional relationship with the gut microbiota [102].

Interestingly, several studies have demonstrated that vitamin D supplementation improved the diversity of gut microbiota in patients with chronic disease, such as cystic fibrosis [103]. Studies also suggested that vitamin D is able to reduce ECM-associated markers in colonic fibroblasts [104]. However, molecular mechanisms underlying the effects of vitamin D on intestinal fibrosis remain obscure.

Amino acids, the building blocks for protein synthesis and their many metabolites, are particularly crucial for immunity and intestinal homeostasis. Tryptophan (TRP) is one of nine essential amino acids and is responsible for modulating the immune system [105]. TRP, along with various TRP metabolites, such as kynurenine and indole, have been extensively studied in the context of IBD. These investigations are warranted, as reduced levels of TRP have been observed in serum from IBD patients [106]. Interestingly, in a clinical study, serum TRP levels were negatively correlated with disease activity in IBD patients [107]. This negative correlation has similarly been observed in DSS-induced colitis in mice and is linked with a decreased level of intestinal commensal *Clostridium sporogenes* [108]. Numerus studies report that the gut microbiota is a driving force mediating the degradation of TRP in the colon. Interestingly, several TRP metabolites, such as kynurenine or indole 3-propionic acid, are produced by metabolic activity of gut microbiota, which in turn modulate host immunity. TRP supplementation reduced TNBS-induced colitis and resulted in increased M2 macrophage polarisation whilst enhancing both mRNA and protein expression of the anti-inflammatory cytokines TGF-β1 and IL-10 [109]. These anti-inflammatory effects are mediated by the activation of the aryl hydrocarbon receptor (AhR). However, few studies have considered the role of the interplay between TRP metabolites and AhR in fibrogenesis. Encouragingly, Monteleone et al., have observed a diminution of ECM-associated proteins in fibroblasts extracted from CD patient strictures following stimulation of AhR by TRP metabolites [110].

Altogether, these observations demonstrate the fundamental role of diet in gut microbiota composition and the activation status of macrophages and thus in the pathogenesis of intestinal inflammation and progression towards its end-stage manifestation, intestinal fibrosis (Figure 2).

## 5. Reprogramming Macrophages for Treatment of Fibrosis: Future Directions

Several methods can be used to reprogram macrophages from traditional approaches, including specific antibody treatments, inhibitors, small molecule drugs allowing the modulation of gene expression modification (i.e., miRNA) and cell therapy. Targeting antibody treatments are the most common method, which target macrophage surface receptors. IgE class antibodies can alter alternatively activated human macrophages towards anti-tumoral functions in cancer, including colorectal cancer [111]. However, because this technique is mostly systemic, it can unfortunately generate many side effects.

Thanks to the emergence of nanotechnology, greatest progress has been made in the design and manufacture of nanoparticles for use in clinical treatments. The use of nanoparticles to target tumor-associated macrophages (TAMs) has been extensively studied. For instance, it has been demonstrated that polymeric nanoparticles, designed to reprogram TAMs, could promote the macrophage switch from the M2 to the M1 phenotype [112]. Thus, delivery-specific cytokines targeting macrophages using nanoparticles might be an attractive way to reprogram macrophages towards anti-fibrotic phenotypes.

Blood and bone marrow-derived macrophages (BMDMs) can be reset ex vivo and transferred to individuals. This might help to alleviate the off-target limitations of directly targeting macrophages in vivo, such as antibodies. For instance, using cell therapy in the clinic is challenging because cells demonstrate a certain plasticity and thus can alter their function in response to environmental changes. Therefore, to maintain polarisation stability, macrophages must be genetically manufactured ex vivo to increase or, on the contrary, tone down the expression of factors associated with macrophage polarisation subsets. Recently, Shields et al., proposed an ex vivo attachment of IFN-γ-loaded phagocytosis-resistant ‘backpacks’ to BMDMs to slowly and continuously deliver IFN-γ in vivo, sustaining pro-inflammatory M1 macrophages in vivo and simultaneously polarising TAMs [113].

Reprogramming of macrophages in the specific context of fibrosis is still poorly studied. In a rodent bleomycin-induced pulmonary fibrosis model, a high expression of sphingosine-1-phosphate receptor-2 (S1PR2) is observed in alveolar macrophages [114]. Interestingly, the knockout of the S1PR2 gene in mice greatly attenuates lung fibrosis, reprograms M2-like fibrosis-inducing macrophages to fibrosis-suppressing macrophages, culminating in dramatic declines in profibrotic cytokine release, including TGF-β1, hydroxyproline biosynthesis, and collagen deposition, with concomitant increases in alveolar airspaces [114]. A clinical trial on ozanimod, a selective S1PR modulator in the treatment of IBD, has been recently published [115]. In this study, an endoscopic remission with mucosal healing was observed among patients who received ozanimod treatment. However, the effect of S1PR inhibition on macrophage status has not been studied and needs to be considered in additional studies as a potential therapeutic candidate in intestinal fibrosis.

Ultimately, it appears to be possible to target metabolic changes in macrophages therapeutically. Approaches might target key enzymes, such as succinate dehydrogenase, which could prevent inflammatory macrophage activation and favour anti-inflammatory pathways, including the expression of IL-10 cytokines. However, therapeutic interventions to reprogram specific M2 subsets through metabolic changes will constitute a major challenge since (i) M2 macrophages exert dual roles in mucosal healing and fibrogenesis and (ii) the metabolic profiles associated with distinct M2 macrophage subtypes need to be clarified. Overall, additional studies are required to explore the therapeutic efficacy of targeting specific metabolic pathways in macrophages. However, as described earlier, a dietary intervention might be considered, targeting macrophage metabolism through the manipulation of systemic nutriment availability. Obviously, dietary approaches also need to consider the effects on gut microbiomes. Both short- and long-term changes in diet can modulate the composition of the gut microbiome, which, in turn, can influence the host immune response through the secretion of several metabolites [116]. Another important consideration is that gut microbiota consume nutriments from the diet and have an impact on their circulating availability which might, in turn, limit the effectiveness of dietary interventions on macrophage polarisation. Thus, how the diet interfaces with the complex interactions between macrophages, their surrounding environment and systemic metabolism needs to be understood before considering dietary interventions on macrophage reprogramming in intestinal fibrosis.

Probiotics as well as living biotherapeutic products are currently considered promising therapeutic options to prevent or treat IBD. Indeed, *Faecalibacterium prausnitzii* (*F. prausnitzii*), which is one of the most abundant commensal bacteria observed in healthy volunteers and is found in lower numbers in IBD patients, appears to be greatly beneficial in fighting intestinal inflammation as well as improving gut barrier function [117,118] and intestinal integrity [119]. Recently, Lenoir et al., have demonstrated that *F. prausnitzii* supernatant containing butyrate could mediate anti-inflammatory responses by promoting the expression of dact3, a gene which negatively regulates the Wnt/JNK pathway [120]. Recently, Wnt signaling pathways has been suggested to be essential in the onset of intestinal fibrosis [121]. Moreover, macrophages (especially those with the M2 phenotype) have been described as a putative source of Wnt ligands and relevant to mucosal integrity [122,123]. Therefore, it is tempting to speculate that probiotics, such as *F. prausnitzii*, may also counteract intestinal fibrosis by acting on macrophage subtypes. However, further investigations are needed to confirm this hypothesis. Recently, a symbiotic combining 12 probiotics with prebiotics (chicory fibres) efficiently reduced both pro-inflammatory cytokines and fibrosis development in mice with DSS-induced colitis. However, additional studies are required to explore the interaction of this symbiotic with macrophage subsets [124].

## 6. Conclusions

Intestinal fibrosis is a major complication in IBD, especially CD, and effective treatments are still lacking. Further studies are required to increase our current knowledge of the mechanisms responsible for intestinal fibrogenesis. Macrophages are the keys actors in mucosal healing; due to their specific plasticity, macrophages can play a dual role, contributing to wound healing processes and/or supporting the development of fibrogenesis. Deciphering macrophage heterogenicity and how heterogenicity influences their interplay with fibroblasts and other intestinal components, such as the gut microbiota, along with immune and stromal cells, is of high value in medical research. Breakthroughs in this topic could allow us to better define the predictive value of macrophages and enlighten us toward the therapeutic potential of macrophage-targeted approaches.

New technological advances, such as single-cell sequencing technologies, have contributed to the identification of new subsets and different phenotypes of intestinal macrophages. During the past decades, the gut microbiota has been subject to extensive investigations, yet the impact of microbiota on intestinal fibrogenesis remains unclear. However, Clooney et al., showed that both microbial diversity and composition were associated with bowel resection in IBD patients and specific microbial signatures can influence the disease course of IBD. Analysis of gut microbiota functions and metabolism has revealed the existence of additional environmental cues capable of influencing macrophage fate and functions. In this context, lifestyle and diet habit, which impact gut microbiota composition, are able to affect the immune system and the phenotypes of macrophages. Nevertheless, the impact of diet on intestinal fibrosis remains largely unexplored and additional studies are needed to fully define the contribution of nutrition and the composition of diet in fibrogenesis. While the transition of macrophages toward an M1 phenotype seems to be a crucial step in the chronicity of inflammation and IBD development, the contribution of macrophages to intestinal fibrosis progression remains controversial, although evidence hints that this process is strongly dependent on M2 macrophage subpopulations. Given the growing interest in macrophages as therapeutic options in treating chronic diseases, it is essential to decipher the mechanisms controlling macrophage switches from a pro-resolving to a pro-fibrotic phenotype and vice versa. Developments in this field could allow us to uncover novel approaches to take advantage of the fine-tuned immunological capability of macrophages and expose unprecedented means of pharmacologically modulating their status to improve the clinical outcomes of patients. Moreover, additional studies are required for deeper understanding of the mechanisms by which specific dietary nutrients promote anti- or pro-fibrotic effects, modulating macrophage activities and the gut microbiota. Overall, given that developments concerning specific treatments for intestinal fibrosis are highly awaited, there is a strong rationale for further research on these topics in the hope of identifying a new class of targets for more efficacious treatment of intestinal fibrosis.

## Figures and Tables

**Figure 1 microorganisms-10-00490-f001:**
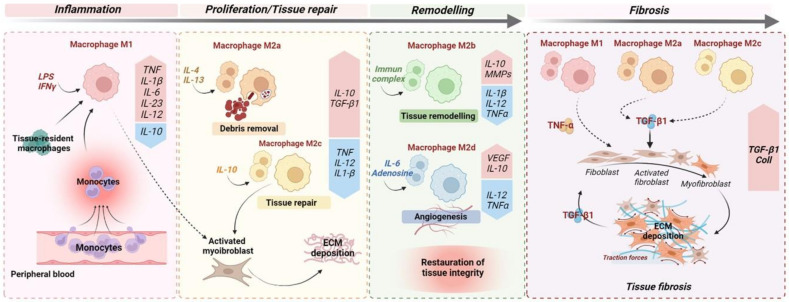
Monocyte and macrophage polarisation during wound healing and fibrosis. The optimal wound healing process could be dived into three distinct phases: inflammation, proliferation, and remodelling. After an acute injury (left), the first phase of wound healing is predominated by pro-inflammatory signals, including lipopolysaccharides (LPSs), that stimulate pro-inflammatory M1 macrophages but also the recruitment upon injury of circulating monocytes, which develop into pro-inflammatory M1 macrophages. In turn, M1 macrophages secrete high concentrations of pro-inflammatory cytokines, such as TNF and IL1-β. In the meantime, IL-4 and IL-13 induce M2a macrophages that clear apoptotic cells. During the proliferative phase, in response to anti-inflammatory cytokines/mediators, including IL-4 and IL-13, and the efferocytosis of apoptotic cells ensured by M2a macrophages, macrophages undergo functional reprogramming toward a pro-restorative phenotype: M2c. These, later, by secreting TGF-β1, promote activation of myofibroblasts and extracellular matrix (ECM) deposition. During the last phase of wound healing, IL-10 is a key anti-inflammatory cytokine produced during the proliferative stage of repair that facilitates tissue remodelling by activating M2b macrophages which release metalloprotease matrix proteins (MMPs) to regulate ECM degradation. In addition, angiogenic response is promoted by M2d macrophage-releasing pro-angiogenic factors, including VEGF. Ultimately, the remodelling phase concludes with complete restoration of tissue. When chronic injuries occur (right), the persistence of M1, M2a, and M2c macrophage activation leads to fibrogenesis through the secretion of pro-inflammatory (TNF) and pro-fibrotic (TGF-β1) cytokines, resulting in sustained myofibroblast activation and leading to excessive ECM deposition.

**Figure 2 microorganisms-10-00490-f002:**
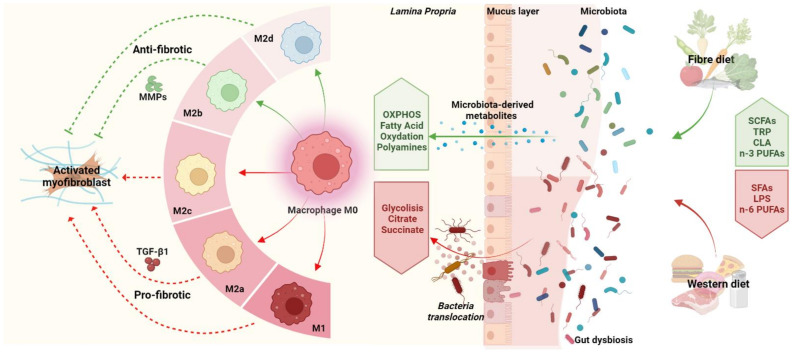
Interplay between diet, gut microbiota, and macrophages during intestinal fibrosis. The gut microbiota regulates intestinal macrophage phenotypes and functions by releasing specific metabolites derived from the metabolism of dietary components. Dietary fibre (green pathway) is metabolised by gut microbiota, leading to short chain fatty acid (SCFA) production, which in turn promotes M2b and M2d macrophage polarisation, which is characterised by an increase of lipid oxidation, providing sustained energy for macrophage anti-fibrotic functions. Dietary components can exert detrimental effects, activating pro-inflammatory and pro-fibrotic processes. The western diet (red pathway), characterised by high fat, high sugar and high salt contents, alters gut microbiota composition (dysbiosis), impairs the gut barrier and leads to n-3 polyunsaturated fatty acid (n-3 PUFA), saturated fatty acid (SFA), and lipopolysaccharide (LPS) production. In turn, macrophages undergo a metabolic reprogramming towards glycolysis and assume an M1, M2a, or M2c phenotype, leading to myofibroblast activation, producing ECM through TGF-β1 secretion.

## Data Availability

Not applicable.

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
