# Peer review of "Gut Microbiota, Macrophages and Diet: An Intriguing New Triangle in Intestinal Fibrosis"

_microorganisms, 2022, doi:10.3390/microorganisms10030490_

Round 1
Reviewer 1 Report
Dear Authors,
After the review process, I have several comments: you should present in the figures legend how was realized; in pages 8 and 9, you should include new data about the significance of vitamins in human well being, e.g. the role of vitamins in neurodegenerative disease, pathologies strictly related with the microbiota dysbiosis; in page 6, the relation between diet and microbiota fingerprint should be expanded, based on the link between obesity, microbiota dysbiosis, and neurodegenerative pathogenesis.
Best regards.
Reviewer 2 Report
Amamou et al discussed an important topic about the link of Gut microbiota, macrophages, and diet in intestinal fibrosis. The authors focused on the following points: a) Macrophages as key player in fibrogenesis. b) Factors that influence macrophage polarization in inflammation and fibrosis: including the Gut microbiota and diet.
c) Interplay between diet, gut microbiota and macrophages in intestinal fibrosis. d) Reprogramming macrophages for treatment of fibrosis: the future directions.
The review is well written and has a good flow. I have some suggestions to improve the quality of the manuscript
1) The cross talk between gut epithelium and macrophage during infection and how this interaction affect intestinal inflammation which is the first step toward the fibrogenesis. This point is missing and it should be included in the review.
2) The role of adaptive immune response especially T -lymphocyte is also missing
3) Therapies that affect gut epithelium and could reduce the risk of IBD and/or its complications.
4) In Diet section, may be the authors mention some probiotics, prebiotics that can be used to manage the IBD.
Round 2
Reviewer 2 Report
The authors addressed my suggestions and the manuscript is improved significantly.